# Long-Term Iron and Vitamin B12 Deficiency Are Present after Bariatric Surgery, despite the Widespread Use of Supplements

**DOI:** 10.3390/ijerph18094541

**Published:** 2021-04-25

**Authors:** Mauro Lombardo, Arianna Franchi, Roberto Biolcati Rinaldi, Gianluca Rizzo, Monica D’Adamo, Valeria Guglielmi, Alfonso Bellia, Elvira Padua, Massimiliano Caprio, Paolo Sbraccia

**Affiliations:** 1Department of Human Sciences and Promotion of the Quality of Life, San Raffaele Open University, 00166 Rome, Italy; arianna.franchi@uniroma5.it (A.F.); roberto.biolcatir@uniroma5.it (R.B.R.); bellia@med.uniroma2.it (A.B.); elvira.padua@uniroma5.it (E.P.); massimiliano.caprio@uniroma5.it (M.C.); 2Department of Systems Medicine, Tor Vergata University, 00133 Rome, Italy; dadamo@med.uniroma2.it (M.D.); valeria.guglielmi@uniroma2.it (V.G.); sbraccia@med.uniroma2.it (P.S.); 3Independent Researcher, Via Venezuela 66, 98121 Messina, Italy; gianlucarizzo@email.it; 4School of Human Movement Sciences, Tor Vergata University, 00133 Rome, Italy; 5Laboratory of Cardiovascular Endocrinology, IRCCS San Raffaele Pisana, 00163 Rome, Italy

**Keywords:** iron, vitamin D, vitamin B12, Roux-en-Y gastric bypass, sleeve gastrectomy, adjustable gastric banding, nutritional deficiency, bariatric surgery

## Abstract

There are few long-term nutritional studies in subjects undergoing bariatric surgery that have assessed weight regain and nutritional deficiencies. In this study, we report data 8 years after surgery on weight loss, use of dietary supplements and deficit of micronutrients in a cohort of patients from five centres in central and northern Italy. The study group consisted of 52 subjects (age: 38.1 ± 10.6 y, 42 females): 16 patients had Roux-en-Y gastric bypass (RYGB), 25 patients had sleeve gastrectomy (SG) and 11 subjects had adjustable gastric banding (AGB). All three bariatric procedures led to sustained weight loss: the average percentage excess weight loss, defined as weight loss divided by excess weight based on ideal body weight, was 60.6% ± 32.3. Despite good adherence to prescribed supplements, 80.7% of subjects (72.7%, AGB; 76.7%, SG; 93.8 %, RYGB) reported at least one nutritional deficiency: iron (F 64.3% vs. M 30%), vitamin B12 (F 16.6% vs. M 10%), calcium (F 33.3% vs. M 0%) and vitamin D (F 38.1% vs. M 60%). Long-term nutritional deficiencies were greater than the general population among men for iron and among women for vitamin B12.

## 1. Introduction

Bariatric surgery is the most effective form of treatment for severe obesity (BMI ≥ 35). Laparoscopic Roux-en-Y gastric bypass (RYGB), adjustable gastric banding (AGB) and laparoscopic sleeve gastrectomy (SG) are the most frequently performed procedures. SG and AGB are restrictive surgical procedures performed to reduce the quantity of food a person can eat. RYGB is a restrictive/malabsorptive operation, which bypasses about 95% of the stomach, the entire duodenum, and 40–150 cm of the jejunum. The risks of nutritional deficiencies are higher in patients undergoing RYGB [1], but in our previous study, we showed that despite the widespread use of supplements, nutritional deficiencies are frequent in patients 5 years after SG [2].

In bariatric patients, recent long-term studies provide evidence of durable 10-year weight loss and improvement in comorbidities and quality of life [3], as well as a reduced risk of developing new health-related comorbidities [4], together with decreased health care utilisation and a drop in direct health care costs [5]. However, many essential long-term health risks of bariatric surgery are still poorly understood. Weight regain (WR) is unfortunately a frequent phenomenon linked to all bariatric procedures [6]. The nutritional requirements in patients who have undergone bariatric surgery are frequently not met despite the widespread use of vitamin–mineral supplements [7]. A recent systematic review revealed that long-term data are needed for RYGB and SG [8]. 

Given this scarcity of long-term data on maintenance of weight loss and nutritional status, we decided to collect the clinical records of subjects undergoing the three most common bariatric surgeries. In this study, we report data 8 years after surgery on weight loss, use of dietary supplements and deficit of some micronutrients in a cohort of patients from five centres in central and northern Italy. We also reviewed data on the prevalence of the same nutritional deficiencies in Italy and Europe for comparison with the study sample.

## 2. Materials and Methods

This was a non-randomised prospective study. Among individuals attending five different centres in central and northern Italy, 60 severely obese subjects undergoing a clinical evaluation before bariatric surgery between 2011 and 2012 were initially included in the present study. 

All procedures were standardised at the different centres. The evaluation consisted of a selected counselling specialties (nutrition, psychiatry, surgery and anaesthesiology) and gastrointestinal endoscopy, as described before [9]. Anthropometric data (weight, height, BMI) were collected at baseline and at each follow-up visit. Following overnight fasting, weight and height were measured while the subjects were wearing only underwear. Outcome measures included absolute (EWL) and percent excess weight loss (%EWL) %EWL = [(Initial Weight) − (Postop Weight)]/[(Initial Weight) − (Ideal Weight)] in which ideal weight is defined by the weight corresponding to a BMI of 25 kg/m^2^. Informed consent was obtained from all individual participants included in the study. The exclusion criteria were as follows: age < 18 years or >65 years, alcoholism, chronic kidney disease (CKD), or taking glucocorticoids, oestrogens or anti-convulsant therapies. All surgical procedures were performed laparoscopically as described elsewhere [2].

Eight patients were excluded from the study because they never answered our communications. Included patients underwent LAGB, SG or RYGB. 

Food habits, caloric intake through food research, level of physical activity and lifestyle, as well as anthropometric data, routine and specific biochemical examinations and vitamin status were assessed. Nutritional follow-up visits provided proper food choices and portion sizes using a visual guide, while also providing education on proper eating behaviour to avoid dumping syndrome, meal patterns and monitoring supplements. An adequate physical activity programme with low intensity aerobic features (at least 150 min/week walking) was also provided. 

### 2.1. Laboratory Data and Diagnostic Criteria

Vitamin and mineral assessments were performed before the surgery and eight years after the operation. The blood plasma samples were taken in the morning (between 7:00 am and 9:00 am). Serum iron was measured using the colorimetric method. Serum 25- hydroxyvitamin D (25OHD) concentrations were determined using an immunoradiometric assay (DiaSorin, Stillwater, MN, USA). Serum vitamin B12 was measured by using the immunoenzymatic method. Measurement of plasma calcium was performed via blood testing with ion-selective electrodes. All of these measurements were performed twice, and the interassay and intraassay coefficients of variation ranged from 1.8 to 9.2%. The deficit limits of serum micronutrients are: iron less than 200 pg/mL, vitamin B12 less than 200 pg/mL, 25OHD less than 25 nmol/L and calcium less than 9 mg/dL.

### 2.2. Nutritional Supplements

Patients were considered adherent if they reported use of nutritional supplements at least 5 days per week for the last month. The following categories of supplements were considered: generic bariatric supplements, RYGB- or SG-specific supplements, multivitamin supplements, vitamin D3 (Cholecalciferol), calcium carbonate, generic vitamin B supplements, and iron (ferrous sulphate). For a complete list of the supplements please refer to the Appendix A.

### 2.3. Data on Prevalence of Nutritional Deficiencies

We searched PubMed, using the following keywords as title/abstract fields: (“deficiency” AND “prevalence”) AND (“vitamin D” OR “vitamin B12” OR “iron” OR “calcium”) AND (“Italy” OR “Europe”). 

We obtained data on the prevalence of iron [10,11,12], 25OHD [13,14] and vitamin B12 [15,16] deficiency. We could not find recent epidemiological data on calcium deficiencies. Deficiency of 25OHD was mostly similar in males and females in European adults. Vitamin B12 deficiency in the UK occurs in about 6% of people under the age of 60. 

### 2.4. Statistical Analysis

Statistical analysis was performed with the SPSS 24.0 software (SPSS, Chicago, IL, USA). Means ± SD with 95% CI or percent proportions were used as descriptive statistics. All quantitative variables were tested for normality distribution using the Kolmogorov–Smirnov test and continuous parameters with non-normal distribution were logarithmically transformed before being used in the subsequent parametric procedures. Differences in continuous variables between groups were assessed using an ANOVA test for multiple comparisons, with the Bonferroni procedure performed as post-hoc analysis. Differences in proportions of discrete traits were assessed using chi-squared test. Within-groups differences in continuous variables between baseline and follow-up values were assessed using the *t*-test for paired data. For all of these analyses, a *p*-value < 0.05, based on a two-sided test, was considered statistically significant.

## 3. Results

The study group consisted of 52 subjects (age: 38.1 ± 10.6 years, 42 females). RYGB was performed in 16 patients, 25 patients had SG and 11 subjects had AGB. All procedures were laparoscopic, there was no operative mortality and revisional bariatric surgery was not performed in any patient. Table 1 shows the main characteristics of subjects before the surgery. 

The mean BMI was 47.5 kg/m^2^; 17.3% of the participants were severely obese (BMI 35–40 kg/m^2^), 48.1% were morbidly obese (BMI 40.1–50 kg/m^2^), 25% were super obese (BMI 50.1–60 kg/m^2^), and 7.7% were super-super obese (BMI over 60.1 kg/m^2^). There were no statistically significant differences in most of the baseline features between the different bariatric surgery types (Table 1). 

All three bariatric procedures led to sustained weight loss (Figure 1). The mean BMI decreased from a baseline of 47.5 kg/m^2^ to 33.3 kg/m^2^ and average weight loss was 41.1 ± 29.3 kg after 96 months. Average percentage excess weight loss (%EWL), defined as weight loss divided by excess weight based on ideal body weight at BMI 25 kg/m^2^, was 60.6% ± 32.3 from baseline to 8 years after surgery. RYGB patients had the greatest weight loss of 50.3 ± 28.8 kg (%EWL = 71.7 ± 27.4%) at the end of follow-up period, although this not dissimilar to that observed in SG patients (45.6 ± 27.8 kg (%EWL = 70.0 ± 21.9%), whereas weight loss in AGB patients was lower (17.6 ± 22.5 kg; %EWL = 22.9 ± 32.3%). (Table 2) Three subjects, all belonging to the AGB group, completely regained their weight loss (WL) over the whole sample.

Table 3 shows the nutritional deficiencies present in the patients stratified by gender. The prevalence of some nutritional deficiencies was higher in females for iron (F 64.3% vs. M 30%), vitamin B12 (F 16.6% vs. M 10%) and calcium (F 33.3% vs. M 0%). Deficiency in 25OHD was more prevalent in males (F 38.1% vs. M 60%). However, these differences were not statistically significant.

A total of 80.8% of subjects (72.7%, AGB; 76.7%, SG; 93.8%, RYGB) reported at least one nutritional deficiency; 21.1% of the subjects had three or more nutritional deficiencies (Table 4). 

The rate of dietary supplement use was similar between SG (88%) and RYGB (93.8%), while it was 72.8% in AGB patients (Table 5). In total, 3.8% of patients with nutritional deficiencies did not use supplements (9.1%, AGB; 4%, SG; 0% RYGB). The improper use of supplements without medical advice or specific nutritional deficiencies was reported in one female in the AGB group and one male and one female in the SG group. The number of supplements taken by patients according to gender and surgery are shown in Appendix A.

## 4. Discussion

Our data confirm that the different bariatric surgery options offer the possibility of weight loss that, although with different amounts of weight regain, is lasting after a period of eight years.

There are few studies in the literature that have evaluated the long-term effects of bariatric surgery on weight regain, nutritional deficiencies and supplement use. Our data on weight loss (%EWL; AGB: 22.9 ± 32.3%, SG: 70.0 ± 21.9%, RYGB: 71.7 ± 27.4%) were similar to those present in the literature. Another paper showed that in AGB patients, the percentage of weight loss was 14.9% at year 7, and mean year 3 to 7 regain was 1.4% of baseline weight [17]. For SG patients, a recent review showed in long-term studies (8+ years of follow-up) a %EWL between 54 and 62.5% [18]. In a 2016 review and meta-analysis on the midterm results of RYGB, the %EWL was 70% in 10 years [19]. In RYGB subjects, a recent trial revealed that a mean percentage of weight loss after 7 years was 28.4%, and between years 3 and 7, there was a mean regain of 3.9% [17]. A total of 405 of 564 patients undergoing RYGB (71.8%) had more than 20% estimated weight loss by year 10 [20]. 

Our data revealed that the maximal weight loss occurred after 12–24 months in the majority of patients. SG and RYGB subjects had a similar weight loss for all eight years with a partial weight regain after the first year in all three surgeries. From the second year onwards, the weight loss remained stable and there was no further WR. On the contrary, in AGB patients there was a partial recovery of the WL (Figure 1). It has been demonstrated that SG generates 30% sustained WL, although a portion of patients had WR in the long term [21]. The mechanism of postoperative WR is not well understood due to the scarcity of long-term data. The easiest explanation may be related to the enlargement of the gastric pouch, which would increase the amount of food eaten per meal. A decrease in ghrelin, a fast-acting hormone that plays a role in meal initiation, may be related to weight loss after bariatric surgery [22], but long-term variations in ghrelin levels do not appear to correlate with WR after the RYGB nadir weight has been achieved [23]. WR may be related to a decrease in the resting metabolic rate after long-term follow-up in bariatric patients [24].

The risks of long-term nutritional deficiencies have been known since the beginning of bariatric surgery [25]. In our sample, patients showed good adherence to the planned integration plan. Often it was necessary to use more than one supplement, especially among patients undergoing restrictive (SG) or restrictive–malabsorptive (RYGB) surgery. However, most patients had at least one nutritional deficiency. In comparison with healthy adults, deficiencies of iron, 25OHD, and vitamin B12 were more frequently reported in our sample of bariatric patients and showed higher incidence (Table 3), without significant differences in proportions between surgical procedures and sex groups, as already reported by previous studies [26].

An important finding was the iron deficiency in males undergoing SG (unfortunately, in our study sample there were no males undergoing RYGB, thus not allowing any comparison between procedures). Data from the literature show that obesity is significantly associated with iron deficiency [27] and long-term energy restriction does not affect iron status [28]. It is therefore necessary to pay attention to the risk of iron deficiency in male patients undergoing bariatric surgery, and irrespective of the technical procedure performed.

Although cholecalciferol supplementation (in the form of vitamin D3 or a multivitamin) was prescribed for the vast majority of patients, the nutritional deficiency of this vitamin was 42.3% in these subjects, particularly (albeit non-significantly) in patients undergoing SG. These data are however in line with the prevalence in the general population [13,14]. Another study showed that, in RYGB subjects, supplementation does not warrant the nutritional status of this vitamin, because after surgery 91% of females and 85% of males had 25OHD deficiency. The study also revealed no change in serum calcium before and after surgery [29]. It is interesting to note that adiposity loss should lead to a release of 25OHD into the bloodstream and increase circulating 25OHD levels with fat mass loss through lifestyle modifications without supplementation [30]. In recent years, several studies have shown that the current limits of blood 25OHD might have been wrongly estimated and that, therefore, widespread nutritional deficiencies in the population may be overestimated [31]. There is no clear evidence for a beneficial effect of cholecalciferol supplementation on cardio-metabolic parameters in obese individuals, and data on such parameters with weight loss are very scarce [32]. Interestingly, our data show that 25OHD deficiency is more prevalent among males than females; this aspect contradicts the lower 25OHD concentrations found in obese females than males, due to the higher fat mass [33].

Vitamin B12 deficiency was found to be higher, mainly in females, compared to the data from the general population, and showed no particular differences between procedures or sex differences. Because vitamin B12 deficiency can, if discovered late, lead to irreversible neurological damage, subjects of both sex who have had bariatric surgery should be supplemented with 1 mg of oral vitamin B12 per day for life [34]. The supplements taken 8 years after surgery were generic multivitamins and minerals, bariatric surgery supplements, supplements specific for a surgery type, or B complex supplements. The use of multivitamins, calcium, iron and 25OHD is lower when compared to another cohort of bariatric patients; however, the use of at least one supplement is higher [35].

There are many limitations to our study. There was a higher prevalence in the sample of females than males; in particular, there was no male patient who underwent RYGB. However, given the small sample size of our study it is possible that such a disproportion might be due to chance. Many important nutritional blood tests, such as albumin, vitamin B1, folic acid and zinc, were not performed. We do not have data on nutritional deficiencies before surgery, so we chose to compare our patients’ data with epidemiological data in the literature.

## 5. Conclusions

Analysis of weight loss in patients undergoing bariatric surgery showed that after 8 years the decrease was similar between patients undergoing SG and RYGB. Patients undergoing AGB lost less weight and experienced greater weight regain. The nutritional deficiencies in bariatric patients, despite the use of new surgery-specific supplements in part of the sample, appear similar to those in other papers published a few years ago. In comparison with the general population, nutritional deficiencies in bariatric patients are greater for iron among males and for vitamin B12 among females.

## Figures and Tables

**Figure 1 ijerph-18-04541-f001:**
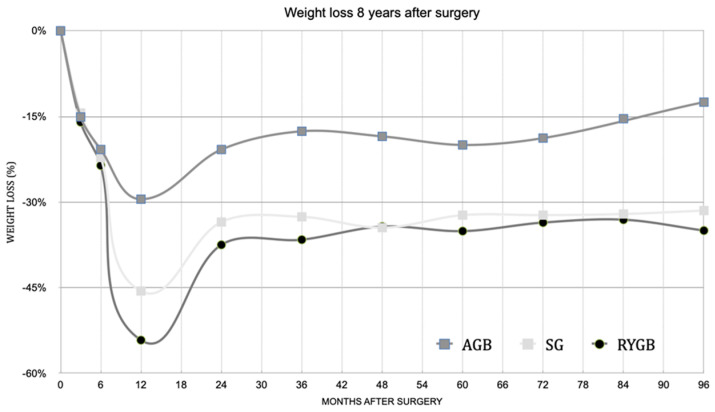
Percent weight loss (%WL) in patients undergoing AGB, SG and RYGB procedures 8 years after surgery. AGB: Laparoscopic adjustable gastric banding SG: Sleeve gastrectomy. RYGB: Roux-en-y gastric bypass. The ANOVA test for multiple comparisons was performed to compare the %WL of the three different surgeries over the 8 years (*p* values: T3; 0.904, T6; 0.850, T12; 0.017, T24; 0.048, T36; 0.001, T48; 0.011; T60; 0.011, T72; 0.013, T84; 0.005; T96; 0.000006). Detailed data are presented in Appendix A.

**Table 1 ijerph-18-04541-t001:** Clinical characteristics of the subjects before the surgery.

(*n*)	AGB (11)	SG (25)	RYGB (16)	*p* *	Total (52)
Age	37.9 ± 10.7	37.7 ± 10.5	38.8 ± 11.1	0.972	38.1 ± 10.6
Sex	M2-F9	M8-F17	M0-F16	0.039	M10-F42
BW	130.6 ± 25.2	136.6 ± 26.3	137.6 ± 33.8	0.816	135.7 ± 29.6
Height (mt)	1.70 ± 0.09	1.66 ± 0.05	1.70 ± 0.08	0.165	
BMI	45 ± 8.5	47.2 ± 7.1	49.8 ± 9.2	0.336	47.5 ± 8.5
BMI 30–34.9 (%)	1 (9)	0	0	0.151	1 (1.9)
BMI 35–40 (%)	3 (27.3)	5 (20)	1 (6.2)	0.324	9 (17.3)
BMI 40.1–50 (%)	4 (36.4)	12 (48)	9 (56.2)	0.597	25 (48.1)
BMI 50.1–60 (%)	3 (27.3)	6 (24)	4 (25)	0.908	13 (25)
BMI > 60.1 (%)	0	2 (8)	2 (12.6)	0.891	4 (7.7)

AGB: Laparoscopic adjustable gastric banding. SG: Sleeve gastrectomy. RYGB: Roux-en-y gastric bypass. BW: Body weight (kg). Data are means ± SD or *n* (%). * ANOVA test for multiple comparisons or chi-square test, as appropriate.

**Table 2 ijerph-18-04541-t002:** Postoperative changes (96 months) in body weight, BMI, and % excess body weight loss.

	T0	(95% CI)	T96	(95% CI)	T0–T96 * *p*	Δ	(95% CI)
	**AGB (11)**						
BW	130.6 ± 25.2	115.7–145.8	113.0 ± 17.9	102.5–125.5	0.03	17.6 ± 22.5	4.3–30.9
BMI	45.0 ± 8.5	39.3–50.6	38.9 ± 5.9	34.9–42.9	0.02	6.0 ± 7.1	1.8–10.2
EBW	57.7 ± 24.3	41.4–74	40.1 ± 16.8	28.8–51.4	0.03		
%EWL			22.9 ± 32.3	1.1–44.6			
	**SG (25)**						
BW	136.6 ± 33.8	123.4–149.9	91.1 ± 15.3	85.1–97.1	<0.001	45.6 ± 27.8	34.7–56.5
BMI	47.2 ± 9.2	43.6–50.8	31.6 ± 4.9	29.7–33.6	<0.001	15.6 ± 8.6	12.2–19.1
EBW	64.5 ± 30.3	52.0–77.1	19.0 ± 14.1	13.1–24.8	<0.001		
%EWL			70.0 ± 21.9	60.9–79.0			
	**RYGB (16)**						
BW	137.6 ± 26.3	124.7–150.5	87.3 ± 16.0	79.5–95.1	<0.001	50.3 ± 28.8	36.2–64.4
BMI	49.8 ± 7.1	46.4–53.3	32.0 ± 6.5	28.8–35.2	<0.001	17.9 ± 9.1	13.4–22.3
EBW	68.9 ± 22.9	57.7–80.1	18.6 ± 17.4	10.0–27.1	<0.001		
%EWL			71.7 ± 27.4	57.1–86.3			
	**ALL SUBJECTS**						
BW	135.7 ± 29.6	127.4–143.9	94.6 ± 18.5	89.4–99.7	<0.001	41.1 ± 29.3	
BMI	47.5 ± 8.5	45.2–49.9	33.3 ± 6.3	31.5–35.0	<0.001	14.3 ± 9.4	
EBW	64.4 ± 26.8	57.0–71.9	23.3 ± 17.7	18.3–28.3	<0.001	41.1 ± 29.3	
%EWL			60.6 ± 32.3	51.6–69.5			

AGB: Laparoscopic adjustable gastric banding SG: Sleeve gastrectomy. RYGB: Roux-en-y gastric bypass. BW: Body weight. EBW = excess body weight to the ideal body weight (BMI = 25 kg/m^2^). %EWL: % weight loss achieved to the ideal body weight. Data are means ± SD and 95% CI. * *t*-test for paired data.

**Table 3 ijerph-18-04541-t003:** Patients’ nutritional deficiencies (%).

%		After 8 Years					Prevalence in Adult Population	References
		AGB (11)	SG (25)	RYGB (16)	*p*	Total (52)		
Iron	F	66.7	58.8	68.8	0.467 *	64.3	8.2–32.9	[10,11,12]
	M	0	37.5	n.a.		30	3	
	Total	54.6	52	68.8	0.555 #	57.7	d.n.a.	
Vitamin D	F	33.3	41.2	37.5	0.667 *	38.1	28–76	[13,14]
	M	50	75	n.a.		60		
	Total	36.3	48	37.5	0.725 #	42.3		
Vitamin B12	F	22.2	17.6	12.5	0.800 *	16.6	6	[15,16]
	M	0	12.5	n.a.		10	6	
	Total	18.2	16	12.5	0.916 #	15.4	6–20	
Calcium	F	22.2	29.4	43.8	0.467 *	33.3	d.n.a.	
	M	0	0	n.a.		0		
	Total	18.2	28	43.8	0.378 #	30.8		

AGB: Laparoscopic adjustable gastric banding. SG: Sleeve gastrectomy. RYGB: Roux-en-y gastric bypass. d.n.a.: data not available. No male subjects underwent RYGB. Nutritional deficiency limits: iron less than 200 picograms/mL, vitamin B12 less than 200 pg/mL. 1,25-dihydroxyvitamin D less than 25 nmol/L, calcium less than 9 mg/dL. * chi-square for differences in proportion of relative nutritional deficiencies between genders. # chi-square for differences in proportion of relative nutritional deficiencies between surgical procedures.

**Table 4 ijerph-18-04541-t004:** Number of nutritional deficiencies in patients by surgery.

Surgery		AGB (11)			SG (25)			RYGB (16)			Total		
%		F (9)	M (2)	Total (11)	F (17)	M (8)	Total (25)	F (16)	M (0)	Total (16)	F (42)	M (10)	Total (52)
Number of deficiencies	0	22.2	50	27.3	29.4	37.5	32	6.3	/	6.2	19	40	23.1
	1	33.3	50	36.4	17.6	12.5	16	50	/	50	33.3	20	30.8
	2	22.2	0	18.2	29.4	25	28	25	/	25	26.2	20	25
	3	22.2	0	18.2	23.5	12.5	20	12.5	/	12.5	19	10	17.3
	4				0	12.5	4	6.3	/	6.3	2.5	10	3.8

AGB: Laparoscopic adjustable gastric banding. SG: Sleeve gastrectomy. RYGB: Roux-en-y gastric bypass. d.n.a.: data not available. No male subjects underwent RYGB (/) (Chi-square test between surgery *p* = 0.424).

**Table 5 ijerph-18-04541-t005:** Patients’ supplement use (%).

%		After 8 Years				
		AGB (11)	SG (25)	RYGB (16)	*p*	Total (52)
Generic Bariatric supplement	F	11.1	47.1	31.3	0.622 *	33.3
	M	0	25	n.a.		20
	Total	9.1	40	31.3	0.180 #	30.8
RYGB or SG specific supplements	F	33.3	17	18.8	0.800 *	26.2
	M	0	12.5	n.a.		10
	Total	27.3	24	18.8	0.865 #	23.1
Multivitamin supplement	F	11.1	5.9	12.5	0.378 *	9.5
	M	50	12.5	n.a.		20
	Total	18.2	8	12.5	0.671 #	11.5
Vitamin D 3 (Cholecalciferol)	F	22.2	23.5	37.5	0.533 *	28.6
	M	50	25	n.a.		30
	Total	27.3	24	37.5	0.643 #	28.8
Calcium carbonate	F	0	0	6.3		2.4
	M	0	0	n.a.		0
	Total	0	0	6.3	0.318 #	1.9
Generic vitamin B supplements	F	0	11.8	12.5		9.5
	M	0	0	n.a.		0
	Total	0	12.5	8	0.487 #	7.7
Iron (ferrous sulphate)	F	33.3	17.6	25	/	23.8
	M	0	0	n.a.		0
	Total	27.3	12	25	0.440 #	19.2

AGB: laparoscopic adjustable gastric banding, SG: sleeve gastrectomy, RYGB: Roux-en-Y gastric bypass. Patients were considered to take the specified supplement if they reported taking that supplement at least 5 days per week for the last month. * chi-square for differences in proportion of relative nutritional deficiencies between genders. # Chi-square for differences in proportion of relative nutritional deficiencies between surgical procedures.

## Data Availability

The data used in this manuscript are publicly available from previous publications and fully disclosed in the Tables of the manuscript.

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
