# Peer review of "Long-Term Iron and Vitamin B12 Deficiency Are Present after Bariatric Surgery, despite the Widespread Use of Supplements"

_ijerph, 2021, doi:10.3390/ijerph18094541_

Round 1

Reviewer 1 Report

1. Some issues in the Materials and Methods section:
a) Page 2 "Serum iron was measured by the colorimetric method and the immunoturbidimetric system, respectively." – some words could have been missed here. Iron was measured by the colorimetric method. What analite was measured by immunoturbidimetric system?
b) Page 3. "25OHD (with or without calcium) containing at least 400 U vitamin D3". Did the patient take calcidiol?  Usually, a simple vitamin D (not hydroxylated at 25 carbon atom) is used as a supplement. Calcidiol is registered as a drug or supplement only in few European countries and is prescribed usually only for certain type of patient (e.g., severe liver diseases).
c) Page 3. "The means ± SD (interquartile range) were used as descriptive statistics for normally distributed or skewed continuous variables, respectively". (interquartile range) should be replaced with "median (interquartile range)"?

2. A serious methodological problem: It is unclear what doses of vitamin D were being taken by the patients after the surgery. "At least 400 IU" (mentioned in the Materials and Methods) means nothing! Were the exact data regarding Vitamin D supplementation being collected? If yes, the reader should be informed about the mean daily dose of vitamin D, also the maximal daily dose. If no, it should be clearly stated in the text, perhaps as one of the limitations.
It seems that many of the patients are taking too low doses of vitamin D. 400 or 600 IU daily doses are very low for obese or overweighted persons, regardless were they operated or not!!! Particularly, if the patients were vitamin D deficient before the surgery (regardless of surgery type), doses 400 or similar are in fact nothing, e.g. equals no supplementation. 

Author Response

  1. Some issues in the Materials and Methods section:

a) Page 2 "Serum iron was measured by the colorimetric method and the immunoturbidimetric system, respectively." – some words could have been missed here. Iron was measured by the colorimetric method. What analite was measured by immunoturbidimetric system?

Reply: In fact, the sentence did not make much sense. We have corrected the sentence as suggested by the reviewer. Serum iron was measured by the colorimetric method.

b) Page 3. "25OHD (with or without calcium) containing at least 400 U vitamin D3". Did the patient take calcidiol?  Usually, a simple vitamin D (not hydroxylated at 25 carbon atom) is used as a supplement. Calcidiol is registered as a drug or supplement only in few European countries and is prescribed usually only for certain type of patient (e.g., severe liver diseases).

Reply: We thank the reviewer for the comment. Vitamin D supplements had cholecalciferol. We edited the text to make this important aspect clearer.

c) Page 3. "The means ± SD (interquartile range) were used as descriptive statistics for normally distributed or skewed continuous variables, respectively". (interquartile range) should be replaced with "median (interquartile range)"?

We thank the reviewer for the comment. We have changed the text as requested.

  1. A serious methodological problem: It is unclear what doses of vitamin D were being taken by the patients after the surgery. "At least 400 IU" (mentioned in the Materials and Methods) means nothing! Were the exact data regarding Vitamin D supplementation being collected? If yes, the reader should be informed about the mean daily dose of vitamin D, also the maximal daily dose. If no, it should be clearly stated in the text, perhaps as one of the limitations.

It seems that many of the patients are taking too low doses of vitamin D. 400 or 600 IU daily doses are very low for obese or overweighted persons, regardless were they operated or not!!! Particularly, if the patients were vitamin D deficient before the surgery (regardless of surgery type), doses 400 or similar are in fact nothing, e.g. equals no supplementation. 

Reply:  We agree with the reviewer. The dosage is indeed too low. 400 IU is the amount contained in the generic supplements which were not intended for people with established vitamin D deficiency. We decided to make things clearer by adding a table summarising the prescribed supplements with micronutrient dosages in the supplementary material. (Table S1).

Table S1. Nutritional characteristics of the supplements

Generic

Bariatric supplements

RYGB specific 

supplements

SG

specific

 supplements

Multivitamin supplements

Vitamin D 3

(Cholecalciferol)

Calcium carbonate 

Generic vitamin B supplements

Commercial Brand

Bariatric © 

WLS “Forte” © 

WLS “Optimum” © 

Multicentrum “Adults” ©

DIBASE ©

Natecal ©

Be-Total Plus ©

Iron

30 mg

VNR% 

214

70 mg

VNR% 

500

28 mg

VNR% 

200

5 mg

VNR% 

36

Vitamin D3

25 mcg

1.000 IU

VNR% 

250

75 mcg

3.000 IU

VNR% 

750

75 mcg

3.000 IU

VNR% 

750

10 mcg

400 IU

VNR% 

100

625  mcg

25.000 IU

VNR% 

6250

Vitamin B12

33 mcg

VNR% 

1320

350 mcg

VNR% 

14000

100 mcg

VNR% 

4000

2.5  mcg

VNR% 

100

1.5  mcg

VNR% 

60

Calcium

162 mg

VNR% 

20

600 mg

VNR% 

75

Reviewer 2 Report

Many thanks for the opportunity to review your manuscript regarding longer-term use of supplements and micronutrient deficiencies in a cohort of bariatric surgery patients.

Please see my queries below in no particular order:

  • final paragraph of intro- please explain what an unselected cohort means.  how were the patients selected?
  • methods - statistical analysis - 2nd sentence may need revising as how data is presented for skewed variables is not reported and the use of 'respectively' in this case seems incorrect. please also ensure that all statistical tests conducted are included in this section as it appears that some are missing (eg gender - chisquared test?).
  • Table 1- were these groups statistically different according to their clinical characteristics?
  • results pg 3 final para final sentence - it states no sig diffs in table 2 - is this meant to say table 1? unclear.
  • Table 2 - change data - unclear what this t-test is comparing between?  not sure what these differences (eg p=0.002, p=0.008) refer to? please ensure included in analysis section of methods also.
  • results pg 4 - the final sentence is not worded clearly  and it's not clear what is significantly different between the different surgery types. please reword for clarity.
  • did anyone regain to baseline weight? if so, how many and which surgery types?
  • Table 3 - please report n & % for clarity, please also report the number of females and males in the table for clarity. chi-squared tests would be able to statistically determine whether deficiencies were higher between men and women - recommend doing these tests and adding to methods analysis section. These recommendations also apply to Table 4.
  • Table 4 - difficult to determine what % of patients without deficiencies don't take at least one supplement? I would also like to know whether those with deficiencies were taking the relevant supplement and how many didn't have deficiencies that were taking supplements (ie were the supps therefore protective?). Additionally, do you have information on dosages and can talk to whether this was high enough to prevent deficiency? how often were these supplements taken?
  • discussion - first para - not sure what WR means - please limit your use of acronyms as this becomes too difficult to read.
  • discussion pg 6 2nd para final sentence - resting metabolic rate?
  • discussion pg 6 3rd para - you really can't say that rates are higher unless statistical tests are done - suggest chi- squared /fishers exact tests.
  • the paragraph in the discussion on vit D (pg 6) is not very clear and could benefit from a reword in order to ensure that your points are clear.
  • please provide information on the dosing of the supplements (and whether this was appropriate for this population) or acknowledge this as a limitation.
  • conclusion - I'm not sure that you can say that weight appears constant - please perform a statistical test to show this. I'm also not sure what surgery-specific supplements refers to  - please provide more details of this in the methods and whether participants were following this?
  • please report in the introduction how your study adds to the body of evidence in this area if there are already longer-term studies reporting weight and nutritional deficiencies. This is currently unclear.
  •  

Author Response

We thank the reviewer for his helpful comments. Indeed, many aspects of the paper needed clarification. We hope we have responded correctly to the comments. We are available for further changes.

  • final paragraph of intro- please explain what an unselected cohort means.  how were the patients selected?

The word "unselected" could be unclear. We have removed it in the introduction and specified the selection criterion among the methods.

  • methods - statistical analysis - 2nd sentence may need revising as how data is presented for skewed variables is not reported and the use of 'respectively' in this case seems incorrect. please also ensure that all statistical tests conducted are included in this section as it appears that some are missing (eg gender - chisquared test?).

Thanks to the reviewer for the very useful comment. We performed Pearson's Chi Square Test for Gender for association between "Gender" and "type of bariatric surgery".

  • Table 1- were these groups statistically different according to their clinical characteristics?

We have added statistical significance. We performed Pearson's Chi Square Test for Gender for association between "Gender" and "type of bariatric surgery".

  • results pg 3 final para final sentence - it states no sig diffs in table 2 - is this meant to say table 1? unclear.

We have corrected the error.

  • Table 2 - change data - unclear what this t-test is comparing between?  not sure what these differences (eg p=0.002, p=0.008) refer to? please ensure included in analysis section of methods also.

The t-test was performed to compare BMI at the beginning and after 8 years. The second t-test was performed to assess the differences between the weight loss in the 3 surgical procedures. We added this information in the analysis section of methods.

  • results pg 4 - the final sentence is not worded clearly  and it's not clear what is significantly different between the different surgery types. please reword for clarity.

  • did anyone regain to baseline weight? if so, how many and which surgery types?

We have added this information to the text “3 subjects in the AGB group completely regained their WL.”

  • Table 3 - please report n & % for clarity, please also report the number of females and males in the table for clarity. chi-squared tests would be able to statistically determine whether deficiencies were higher between men and women - recommend doing these tests and adding to methods analysis section. These recommendations also apply to Table 4.

We have rearranged table 3 and 4 by inserting the required data. We added a new table (s2) in the supplementary material showing data on the number of supplements taken stratrified by surgery and gender. Statistical analysis did not report significant data. In the description of the table the p values were reported.

  • Table 4 - difficult to determine what % of patients without deficiencies don't take at least one supplement? I would also like to know whether those with deficiencies were taking the relevant supplement and how many didn't have deficiencies that were taking supplements (ie were the supps therefore protective?). 

Unfortunately, we were unable to obtain this information from the database.

  • Additionally, do you have information on dosages and can talk to whether this was high enough to prevent deficiency? how often were these supplements taken?

Information on dosage  has been added to table S1 in the supplementary material. Patients were considered to take the specified supplement if they reported taking that supplement at least 5 days per week.

  • discussion - first para - not sure what WR means - please limit your use of acronyms as this becomes too difficult to read. discussion pg 6 2nd para final sentence - resting metabolic rate?

We have revised the first part of the discussion and fixed the error pointed out by the reviewer.

  • discussion pg 6 3rd para - you really can't say that rates are higher unless statistical tests are done - suggest chi- squared /fishers exact tests.

The ANOVA test was performed to compare the %WL of the three different surgeries over the 8 years.

  • the paragraph in the discussion on vit D (pg 6) is not very clear and could benefit from a reword in order to ensure that your points are clear.please provide information on the dosing of the supplements (and whether this was appropriate for this population) or acknowledge this as a limitation.

We agree with the reviewer. The dosage is indeed too low. 400 IU is the amount contained in the generic supplements which were not intended for people with established vitamin D deficiency. We decided to make things clearer by adding a table summarising the prescribed supplements with micronutrient dosages in the supplementary material. (Table S1).

  • conclusion - I'm not sure that you can say that weight appears constant - please perform a statistical test to show this. 

The ANOVA test was performed to compare the %WL of the three different surgeries over the 8 years.

  • I'm also not sure what surgery-specific supplements refers to  - please provide more details of this in the methods and whether participants were following this?

All information on the new type-specific supplements has been added to table S1 in the supplementary material.

  • please report in the introduction how your study adds to the body of evidence in this area if there are already longer-term studies reporting weight and nutritional deficiencies. This is currently unclear.

As we wrote in the introduction, there are not many studies on the subject. We also reported two reviews stating that long-term data are needed for RYGB and LSG. We added the following sentences in the introduction. “Given this scarcity of long-term data on maintenance of weight loss and nutritional status, we decided to collect clinical records of subjects undergoing the three most common bariatric surgeries.”

Round 2

Reviewer 1 Report

No suggestions

Author Response

Thank you!

Reviewer 2 Report

Many thanks again for your changes made to the manuscript. This has improved the clarity of the manuscript.

Please see additional edits below:

  • LSG and LG used in abstract and I think they represent the same thing? please check throughout as (for eg) SG used in supp table 1 and results.
  • methods - para 1 - severely obese rather than severe obese?
  • methods para 4- should follow up visits be mentioned at the start of this paragraph rather than the end of the previous one?
  • supp table 1 - what is a cp? and is this the recommended amount to be taken or what was taken?
  • statistical analysis - what does %WL mean? what tests were done for non-parametric data?please either list all tests performed for each variable or summarise all types of tests and spell out what specific tests were done in the tables for each variable.
  • table 1- given there are 3 surgical groups - where do the differences lay in gender between the surgeries?  there also appear to be diffs btw BMI categories - why not assessed? (for eg, using fishers exact test) why do you think proportionately more males had sleeve gastrectomy?
  • tables - please also ensure that any missing data is mentioned in the tables.
  • table 2 - please spell out where the differences exist between the 3 surgical groups using post-hoc analysis. eg SG and RYBG appear similar but both are different to AGB - this is mentioned in the results section but not clear in the table. tables should stand alone. 
  • table 2-  why are there no stat tests comparing EBW & %EWL between surgical groups? recommend adding in.
  • thank you for clarifying who completely regained weight lost. please make it clear that this was those 3 subjects only out of the entire sample.
  • figure 1 - thank you for adding in the statistical analysis. please make it clear where the diffs lay between the groups at these time points. I would suspect SG and RYGB aren't different at these timepoints.  Seems strange that at 12 months, there isn't a significant difference between surgeries given this is actually the largest difference between the groups? why is this not significant but all other timepoints after this are? Is there missing data that becomes important? please consider adding in any sample sizes to reflect missing data.
  • My previous comment was - Table 4 - difficult to determine what % of patients without deficiencies don't take at least one supplement? I would also like to know whether those with deficiencies were taking
    the relevant supplement and how many didn't have deficiencies that were taking supplements (ie were the supps therefore protective?).  Your response was that this information wasn't available from the database. Please explain why and if it isn't available please note this as a limitation in your discussion.
  • interpretation of table 3 - please note that if your p value isn't significantly different, that means that there are no diffs between groups (unless you have a power issue with your sample sizes?). Please reword. Something can't be higher but have a non-significant p value.  Similarly in the discussion, you say a higher prevalence of iron deficiency in rygb - please review (albeit p value was close to significant for iron so it could show a trend for a difference between surgeries).
  • table 3 - what does the ? mean? If data isn't available at the population level, please write this.  what does / mean? please ensure tables stand alone. same for table 4 and 5.
  • table 4 & results - your table 4 suggests that 21.1% of subjects had 3 or more deficiencies whereas your results section says 23.1%? 80.7% reporting no deficiencies appears incorrect also when looking at table 4. your breakdown by surgery in the results section appears incorrect also. please review.
  • table 5 - did supplement use differ by surgery? recommend a chisquare test to be run to determine this. if the test compares surgeries, why can't a p value be shown for iron supplement?
  • table 5 - please make it clear what the p value represents and what test this was. I would suspect it was by surgery (and not by gender) but I can't be certain.  what period of time did they need to be taking the supplement for? eg 5 days per week for the last  week/ month/  year? Please specify in the methods.
  • please comment on whether these supplements are adequate for the surgery undertaken.
  • in your abstract you mention that average weight loss was constant in rygb and sg from the third year after surgery. how have you statistically proved this? Is this a repeated measures analysis finding? OR have you done paired t-tests on weight loss for each different surgery between the years and showed no significant p value? given this is a major finding that you mention in your conclusion and abstract, I would recommend a statistical test to show this.
  • discussion - what is r weight regain? mentioned in first and 2nd paragraphs.
  • it would be valuable if figure 1 included in the std deviations in the data at each time point? are they narrow or wide?
  • you mention sex in the discussion but gender throughout. please use term consistently as these usually mean different things.
  • please consider revision of your conclusion in line with suggestions made above.

Author Response

Dear editors and reviewers,

First of all, we would like to thank you for the valuable impulses that have led us to profoundly change the paper. Hoping that we have satisfied your requests as much as possible, we ask you to re-evaluate our paper.

The Authors

Point-by-point answers to the reviewer's requests:

  • LSG and LG used in abstract and I think they represent the same thing? please check throughout as (for eg) SG used in supp table 1 and results.

Corrected

  • methods - para 1 - severely obese rather than severe obese?

Corrected

  • methods para 4- should follow up visits be mentioned at the start of this paragraph rather than the end of the previous one?

Corrected

  • supp table 1 - what is a cp? and is this the recommended amount to be taken or what was taken?

We have corrected the table to include the amount of supplements actually taken by the subjects.

  • statistical analysis - what does %WL mean?

Percent Weight loss (%WL). Added to the text.

  • what tests were done for non-parametric data?

All the quantitative variables were first tested for normality distribution with the Kolmogorov-Smirnov test. Non-parametric variables (i.e. with a non-normal distribution) were therefore logarithmically transformed before being used in the subsequent parametric procedures. This point is now better specified in the paragraph of the Methods section regarding statistical analysis.

  • please either list all tests performed for each variable or summarise all types of tests and spell out what specific tests were done in the tables for each variable.

The paragraph regarding statistical analysis has been reworded and now includes all tests performed to analyze data (both quantitative and discrete traits). Tables were also modified accordingly and now include relative tests used for each variable.

  • table 1- given there are 3 surgical groups - where do the differences lay in gender between the surgeries?  there also appear to be diffs btw BMI categories - why not assessed? (for eg, using fishers exact test) why do you think proportionately more males had sleeve gastrectomy?

Fisher’s exact test was performed to assess differences in initial BMI categories between surgical procedures (now displayed in Table 1 as requested). No significant differences between procedures were found. We agree with the Reviewer that the major proportion of males in the SG group, compared with the other two, is a limitation of this study. We reinforced this concept in the Discussion section.

  • tables - please also ensure that any missing data is mentioned in the tables.

Added

  • table 2 - please spell out where the differences exist between the 3 surgical groups using post-hoc analysis. eg SG and RYBG appear similar but both are different to AGB - this is mentioned in the results section but not clear in the table. tables should stand alone.

We removed p-values related to multiple comparisons from Table 2, which now displays only differences in the selected quantitative parameters between baseline (T0) and the end of follow-up (T96), stratified by surgical procedure. Post-hoc multiple comparisons between procedures are now reported only in the text.

  • table 2-  why are there no stat tests comparing EBW & %EWL between surgical groups? recommend adding in.

According to the previous point, in order to simplify the meaning of the table, we decided to remove multiple comparisons from Table 2, reporting these analysis only in the text.

  • thank you for clarifying who completely regained weight lost. please make it clear that this was those 3 subjects only out of the entire sample.

Corrected

  • figure 1 - thank you for adding in the statistical analysis. please make it clear where the diffs lay between the groups at these time points. I would suspect SG and RYGB aren't different at these timepoints.  Seems strange that at 12 months, there isn't a significant difference between surgeries given this is actually the largest difference between the groups? why is this not significant but all other timepoints after this are? Is there missing data that becomes important? please consider adding in any sample sizes to reflect missing data.

The reviewer's observation is absolutely pertinent. Unfortunately there was a typo in the p referring to T12. It is actually 0.017 and not 0.097. We have corrected this error and included all the required information in Table S2 in the supplementary material.

  • My previous comment was - Table 4 - difficult to determine what % of patients without deficiencies don't take at least one supplement? I would also like to know whether those with deficiencies were taking the relevant supplement and how many didn't have deficiencies that were taking supplements (ie were the supps therefore protective?).  Your response was that this information wasn't available from the database. Please explain why and if it isn't available please note this as a limitation in your discussion.

We apologise but we did not understand the previous request. 3.8% of patients with nutritional deficiencies did not use supplements (9.1%, AGB; 4%, SG; 0% RYGB). Improper use of supplements without medical advice or specific nutritional deficiencies was reported in one female in the AGB group and one male and one female in the SG group. The number of supplements taken by patients according to gender and surgery are shown in table 3S in the supplementary material.

  • interpretation of table 3 - please note that if your p value isn't significantly different, that means that there are no diffs between groups (unless you have a power issue with your sample sizes?). Please reword. Something can't be higher but have a non-significant p value.  Similarly in the discussion, you say a higher prevalence of iron deficiency in rygb - please review (albeit p value was close to significant for iron so it could show a trend for a difference between surgeries).

The discussion section has been reworded making clear that no significant differences occurred in percent proportions of nutritional deficiencies according to surgical procedures. The manuscript was modified as follows: “In comparison with healthy adults, deficiency of iron, 25OHD and vitamin B 12 were more frequently reported in our sample of bariatric patients (table n.3), without significant differences in proportions between surgical procedures and gender groups, as already reported by previous studies [26]. An important finding was the iron deficiency in males undergoing SG (unfortunately, in our study sample there were no males undergoing RYGB, thus not allowing any comparison between procedures). Data from the literature show that obesity is significantly associated with iron deficiency [27] and long-term energy restriction does not affect iron status [28]. It is necessary therefore to pay attention to the risk of iron deficiency in male patients undergoing bariatric surgery, and irrespective of the technical procedure performed.”

  • table 3 - what does the ? mean? If data isn't available at the population level, please write this.  what does / mean? please ensure tables stand alone. same for table 4 and 5.

We have corrected the tables as requested

  • table 4 & results - your table 4 suggests that 21.1% of subjects had 3 or more deficiencies whereas your results section says 23.1%? 80.7% reporting no deficiencies appears incorrect also when looking at table 4. your breakdown by surgery in the results section appears incorrect also. please review.

We have corrected the table as requested

  • table 5 - did supplement use differ by surgery? recommend a chisquare test to be run to determine this.

The differences between procedures for the total number of patients were assessed with a chi-square test.(#) No significant difference between the procedures.

  • if the test compares surgeries, why can't a p value be shown for iron supplement?

We added a chi-square test. No significant difference between the procedures.

  • table 5 - please make it clear what the p value represents and what test this was. I would suspect it was by surgery (and not by gender) but I can't be certain. 

We apologise, the table was indeed not very clear. We have included some information in the caption of the table to make it clearer. For each supplement a t-test was performed to assess gender differences.(*) The differences between procedures for the total number of patients were assessed with a chi-square test.(#)

  • what period of time did they need to be taking the supplement for? eg 5 days per week for the last  week/ month/  year? Please specify in the methods.

Patients were considered adherent if they reported use of nutritional supplements at least 5 days per week for the last month. We have added this information to the text

  • please comment on whether these supplements are adequate for the surgery undertaken.

We have added a comment in the discussion.

  • in your abstract you mention that average weight loss was constant in rygb and sg from the third year after surgery. how have you statistically proved this? Is this a repeated measures analysis finding? OR have you done paired t-tests on weight loss for each different surgery between the years and showed no significant p value? given this is a major finding that you mention in your conclusion and abstract, I would recommend a statistical test to show this.

Data are presented in table 2S in the supplementary material. The ANOVA test was performed to compare the %WL of the three different surgeries over the 8 years (*). T-test was performed to compare SG and RYGB (#). We rewrote part of the text.

  • discussion - what is r weight regain? mentioned in first and 2nd paragraphs.

Corrected

  • it would be valuable if figure 1 included in the std deviations in the data at each time point?

We have added a table (S2) in the supplementary material showing the standard deviations and the 95% Confidence Interval of the Mean (Table S2. Percent Weight loss (%WL) in patients undergoing AGB, SG and RYGB procedures 8 years after surgery.)

  • you mention sex in the discussion but gender throughout. please use term consistently as these usually mean different things.

Corrected

  • please consider revision of your conclusion in line with suggestions made above.

We have rewritten part of the conclusion as suggested by the reviewer.